# Functional analysis of Cdc20 reveals a critical role of CRY box in mitotic checkpoint signaling

Yuqing Zhang[1], Rose Young [2], Dimitriya H. Garvanska[3], Chunlin Song[1], Yujing Zhai[4], Ying Wang[4], Hongfei Jiang [1], Jing Fang[1], Jakob Nilsson [3], Claudio Alfieri [2✉] & Gang Zhang [1✉]

Accurate mitosis is coordinated by the spindle assembly checkpoint (SAC) through the mitotic checkpoint complex (MCC), which inhibits the anaphase-promoting complex or cyclosome (APC/C). As an essential regulator, Cdc20 promotes mitotic exit through activating APC/C and monitors kinetochore-microtubule attachment through activating SAC. Cdc20 requires multiple interactions with APC/C and MCC subunits to elicit these functions. Functionally assessing these interactions within cells requires efficient depletion of endogenous Cdc20, which is highly difficult to achieve by RNA interference (RNAi). Here we generated Cdc20 RNAi-sensitive cell lines which display a penetrant metaphase arrest by a single RNAi treatment. In this null background, we accurately measured the contribution of each known motif of Cdc20 on APC/C and SAC activation. The CRY box, a previously identified degron, was found critical for SAC by promoting MCC formation and its interaction with APC/C. These data reveal additional regulation within the SAC and establish a novel method to interrogate Cdc20.

[1] Cancer Institute, The Affiliated Hospital of Qingdao University, Qingdao University, Qingdao, China. [2] Chester Beatty Laboratories, Structural Biology Division, Institute of Cancer Research, London, UK. [3] The NNF Center for Protein Research, University of Copenhagen, Copenhagen, Denmark. [4] School of Public Health, Qingdao University, Qingdao, China. ✉email: claudio.alfieri@icr.ac.uk; zhanggang_sma@qdu.edu.cn

Correct segregation of replicated chromatids during mitosis is required for the maintenance of intact genetic information. Disturbance of the mitotic process can lead to serious consequences like abortion, aneuploidy-related syndromes, or cancer. Cooperation between the SAC and APC/C ensures correct and timely mitotic progession[1]. Cdc20 is a key subunit of the MCC[2,3], which is produced at unattached kinetochores by the SAC signaling pathway. The MCC binds and inhibits the APC/C E3 ligase to delay the transition from metaphase to anaphase, thus providing sufficient time to establish stable attachment between kinetochores and microtubules[2]. Cdc20 is the mitotic co-activator of the APC/C. Upon APC/C binding, Cdc20 promotes conformational changes on the APC/C catalytic module that favor the loading of the E2 ligases UBE2C and UBE2S, which brings activated ubiquitin ready for transfer to an APC/C-bound substrate[4]. Besides activating the APC/C, Cdc20 recognizes APC/C-targeting sequences or degrons on substrates and facilitates substrate ubiquitination by APC/C[5]. Two Cdc20 molecules are part of the APC/C$^{MCC}$ complex, one in the MCC and the other in the APC/C[6–8]. The binding of the MCC to the APC/C interferes with Ube2C binding[7]. Moreover, the MCC component BubR1 interacts with Cdc20$^{APC/C}$ and blocks substrate recognition. Thus, MCC directly inhibits APC/C from ubiquitinating Cyclin B1 and securin, thereby causing a mitotic arrest[9,10].

Cdc20 has an unstructured N-terminal region (residues 1–173) and a seven-bladed β-propeller or WD40 domain at the C-terminus (residues 174–470). The N-terminal region contains motifs that bind: Mad1 (BM1), Mad2 (KILR), Apc8 (C box; KILR), and two degrons recognized by APC/C$^{Cdh1}$ (the KEN box and CRY box)[11–17]. There are also numerous residues phosphorylated by the mitotic kinases Cdk1, Plk1, and Bub1[18–21]. The β-propeller domain contains binding pockets for the D box and KEN box degrons and other APC/C-recognition sequences (e.g., ABBA motif)[22–24]. At the C-terminal end, an IR motif mediates the interaction with Apc3 (for Cdc20$^{APC/C}$) or Apc8A (for Cdc20$^{MCC}$)[7,12,14].

Similar to other mitotic checkpoint proteins like Bub1 or Mad1, Cdc20 is extremely difficult to deplete in an efficient manner by RNAi to block mitotic progression. It is estimated that the endogenous Cdc20 protein level has to be reduced to less than 5% for the cells to show mitotic arrest[25,26]. On the other hand, complete inactivation of Cdc20 using the gene trap method results in metaphase arrest followed by apoptosis[27]. Thus, precisely measuring the contribution of each Cdc20 functional motif to APC/C and SAC activation in a clean null background remains a technical challenge.

Previously, we generated a *Bub1* knockout cell line by CRISPR/Cas9. These cells are viable due to residual Bub1 protein and are highly sensitive towards Bub1 RNAi[28]. Using this cell line, we completely removed the residual Bub1 by RNAi and revealed the distinct roles of Bub1-Mad1 and RZZ-Mad1 on SAC activation as well as the biological significance of separating the kinase activity and phosphatase activity within the Bub1 complex[28,29]. Here, we applied a similar strategy for the study of Cdc20 and precisely measured the contribution of each motif to the activation of SAC and APC/C. Though most of our results are in line with the previous studies, we found that the disruption of the APC/C binding motifs individually on Cdc20 completely abolished the activation of APC/C in cells, which had not been demonstrated before. Furthermore, we discovered that the cryptic degron CRY box was critical for SAC activation but not for APC/C activation. By structural and functional analysis, we show that the CRY box of Cdc20$^{MCC}$ forms multiple interactions at the interface between MCC and the APC/C to facilitate MCC formation and MCC-mediated APC/C inhibition. Due to the high similarity between the C box and the CRY box, the core sequences can be exchanged while maintaining the function of each motif. The arginine within the CRY box is highly conserved, indicating an essential role in the SAC across eukaryotic cells.

## Results

### Generation of *Cdc20* RNAi-sensitive cell line.
To achieve a clean background and avoid cell death due to the complete loss of Cdc20, we took a similar strategy as utilized previously for Bub1[28,29] (Supplementary Fig. 1a). We designed three single-guide RNAs (sgRNAs) targeting distinct positions on exon 1 and 2 and generated many cell lines with either low or undetectable Cdc20 by CRISPR/Cas9 (Supplementary Fig. 1b–e, Fig. 1a). Since complete inactivation of *Cdc20* causes metaphase arrest and apoptosis[27], we reasoned that the survival of the knockout cells without detectable Cdc20 is very likely supported by residual Cdc20 protein not detected by western blot. Indeed, Cdc20 peptides were detected by mass spectrometry from the immunoprecipitate of clone KO3-9, with an antibody against the C-terminus of Cdc20 (Supplementary Fig. 1f). All these clones spent additional 30–70 min in mitosis compared to parental cells (Fig. 1b). More importantly, 24 h after treatment with a small interfering RNA (siRNA) oligo against Cdc20, the majority of mitotic cells from such clones were arrested at metaphase till cell death while parental cells were briefly delayed before entering anaphase (Fig. 1c). The metaphase arrest caused by RNAi in knockout cells was fully rescued by reintroducing RNAi resistant YFP-Cdc20 (Fig. 2c–f).

In conclusion, the above results show the successful generation of Cdc20 RNAi-sensitive cell lines. Since the clones from sgRNA #3 gave more homogenous results, we decided to characterize these cell lines further.

### *Cdc20* knockout cell lines are unable to activate the SAC.
We first examined the mitotic progression by live cell imaging using a fluorescent histone marker. The results showed that the prolonged mitosis in the knockout cells was due to a metaphase-anaphase transition delay (Supplementary Fig. 2a, b). The degradation of YFP-Cyclin B1 was much slower in the knockout cells than in parental HeLa cells, indicating the delay was caused by inefficient Cyclin B1 degradation (Supplementary Fig. 2c, d). The slow Cyclin B1 degradation could be a result of less active APC/C$^{Cdc20}$ or increased SAC strength. To discriminate between these two possibilities, we examined SAC strength in the knockout cells. A low dose of nocodazole partially depolymerized microtubules and activated the mitotic checkpoint, which arrested the parental HeLa cells in mitosis with a median time of around 585 min. Interestingly, none of the three knockout cell lines were arrested in mitosis after nocodazole treatment (Fig. 1d). Similar results were obtained with paclitaxel, another microtubule toxin that dampened microtubule dynamics and activated mitotic checkpoint (Fig. 1e). When the knockout cells were treated with reversin, an inhibitor of the checkpoint kinase Mps1[30], the mitotic length was not affected while in the parental cells, accelerated mitosis was observed (Fig. 1f). The SAC was fully restored when the knockout cells were supplemented with YFP-Cdc20 indicating the checkpoint defect was caused by the low levels of Cdc20 (Fig. 1g).

In summary, the characterization of the *Cdc20* knockout cell line indicates that the residual Cdc20 can partially activate APC/C but not the SAC.

### Functional analysis of Cdc20 motifs in the Cdc20 null background.
After characterizing the *Cdc20* RNAi-sensitive cell lines, we decided to quantify the contribution of each documented

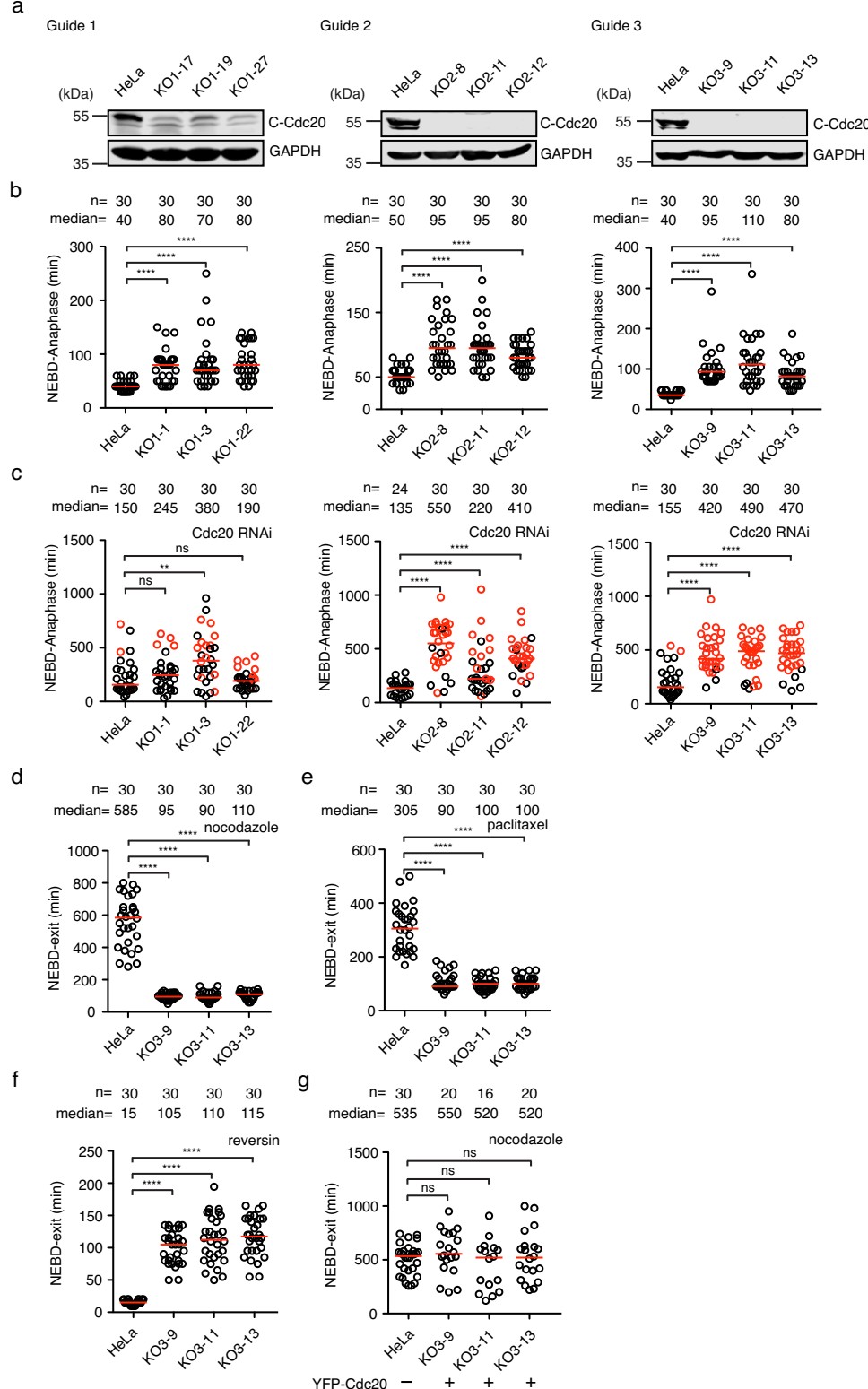

**Fig. 1 Generation of RNAi-sensitive *CDC20* knockout cell lines by CRISPR/Cas9. a** Cdc20 in knockout cells from three sgRNAs was examined by western blot with an antibody targeting the Cdc20 C-terminal region. See Supplementary Fig. 5 for uncropped blots. **b–g** The time from NEBD (nuclear envelope breakdown) to anaphase for unperturbed mitosis or NEBD to mitotic exit after microtubule toxin treatment. Each circle represents the time of a single cell, and the red line indicates the median time. The number of cells analyzed per condition is indicated above ($n = $ X). A representative experiment of two independent experiments is shown. Mann–Whitney *U*-test was applied. ns means not significant; ** means $P < 0.01$; **** means $P < 0.0001$. **b** The mitotic time of the knockout cells. **c** The mitotic time of the knockout cells after depleting the residual endogenous Cdc20 by RNAi. Red dots are the cells that died after the metaphase arrest. **d** The mitotic time of cells treated with nocodazole (30 ng/ml). **e** The mitotic time of cells treated with paclitaxel (0.1 μM). **f** The mitotic time of cells treated with reversin (0.5 μM). **g** The mitotic time of knockout cells transfected with YFP-Cdc20 expression plasmid and treated with nocodazole (30 ng/ml).

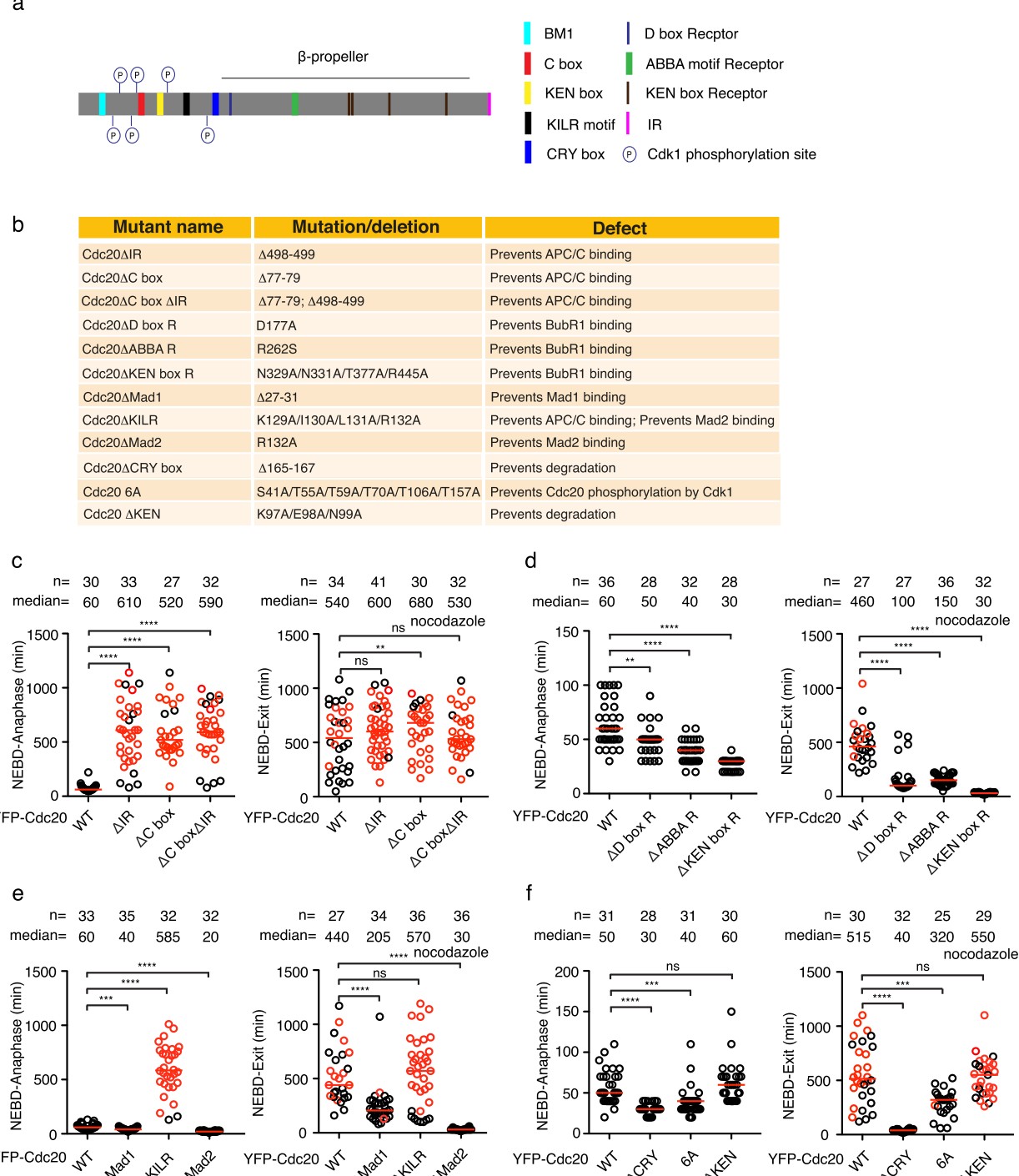

**Fig. 2 Functional analysis of Cdc20 in the null background. a** Schematic of the positions of the motifs or phosphorylated sites analyzed in this study. **b** Mutation details and known functions of the motifs or phosphorylated sites in this study. **c–f** The time from NEBD to anaphase (left) or from NEBD to mitotic exit in the presence of nocodazole (right, 30 ng/ml) of the knockout cells complemented with wild-type or mutant YFP-Cdc20 and depleted the residual endogenous Cdc20 by RNAi. Each circle represents the time of a single cell, and the red line indicates the median time. The red circle means the cell died after mitotic arrest. The number of cells analyzed per condition is indicated above (n = X). A representative experiment of two independent experiments is shown. Mann–Whitney U-test was applied. ns means not significant. ** means P < 0.01;*** means P < 0.001; **** means P < 0.0001.

Cdc20 interaction to the SAC and APC/C activation. A series of Cdc20 mutants were generated according to previous studies (Fig. 2a, b). Localization analysis by immunofluorescence showed all the mutants except Cdc20ΔABBA R, the one lacking the ABBA-binding motif decorated kinetochores (Supplementary Fig. 3a). The knockout cell line 3–9 was used for the functional analysis of these mutants. KO3–9 cells were synchronized by double thymidine, and the RNAi-resistant plasmids expressing YFP-Cdc20 variants were transfected after the first thymidine arrest. RNAi against Cdc20 was conducted the next day during the second thymidine arrest, and live cell imaging was performed 24 h later. Unperturbed mitosis and nocodazole-challenged mitosis were recorded side by side for each mutant to examine the activity of APC/C and SAC separately.

For unperturbed mitosis, removal of the C-terminal IR motif or the C box individually or combinatorially, which compromises Cdc20 binding to the APC/C, caused metaphase arrest followed by cell death. Similar results were obtained when the cells were treated with nocodazole (Fig. 2c). To confirm that the arrest was caused by failed activation of APC/C but not by a highly active SAC, we lowered the concentration of nocodazole and recorded the time the cells spent in mitosis again. The cells complemented with wild-type Cdc20 exited mitosis significantly faster with nocodazole at 15 ng/ml than at 30 ng/ml (median time 430 min vs 570 min). In contrast, cells expressing Cdc20 lacking APC/C interaction motifs were arrested for the same amount of time irrespective of nocodazole concentration changes (Supplementary Fig. 3b). Thus, the arrest observed with these mutants is due to their lack of APC/C activation.

The Cdc20 mutants defective at BubR1 binding showed accelerated mitosis in both unperturbed mitosis and SAC-activated mitosis. Cdc20ΔKEN box R gave the strongest SAC defect, followed by Cdc20ΔD box R and Cdc20ΔABBA R (Fig. 2d). Similarly, both Mad1 and Mad2 binding-defective Cdc20 mutants had fast mitosis in the absence or presence of nocodazole. Cdc20ΔMad2 showed an SAC defect as strong as Cdc20ΔKEN box R, while Cdc20ΔMad1 displayed a mild SAC defect (Fig. 2e). Another mutant Cdc20ΔKILR defective at binding with both APC8 and Mad2 exhibited similar mitotic arrest and cell death in unperturbed mitosis and nocodazole-treated mitosis as Cdc20ΔIR or Cdc20ΔC box indicating its critical role in APC/C activation. We noticed a small pool of cells expressing Cdc20ΔKILR exit mitosis quickly when challenged with nocodazole (Fig. 2e). We also examined the effect of Cdc20 phosphorylation by Cdk1 which has been reported to lower APC/C activity. In line with the previous studies, Cdc20 6 A mutant induced accelerated mitosis in the absence and presence of nocodazole (Fig. 2f). Cdc20ΔKEN mutant did not induce a significantly accelerated or delayed mitosis in either unperturbed mitosis or SAC-activated mitosis arguing that it is not critical for mitosis (Fig. 2f).

The most unexpected phenotype comes from mutating the CRY box, an unconventional degron that is poorly characterized in human somatic cells. In our system, Cdc20ΔCRY accelerated unperturbed mitosis, which was due to an abolished SAC, as Cdc20ΔCRY complemented cells were not arrested in the presence of nocodazole (Fig. 2f).

Given this surprising result, we decided to investigate the CRY box further.

**CRY box is required for SAC via multiple interactions with MCC and APC/C.** In order to understand the molecular mechanism of how the CRY box participates in SAC-dependent inhibition of the APC/C, we performed a structure-function analysis of this motif based on the APC/C$^{MCC}$ structure determined by cryo-electron microscopy (cryo-EM)[7,31] (Fig. 3a, b). From this structure, the Cdc20$^{MCC}$ CRY box (residues 162-170) forms a loop that precedes the first WD40 repeat (Fig. 3c, d). Two basic residues preceding the core CRY sequence, Cdc20$^{MCC}$ R162 and K163, are in proximity to two acidic residues, E180 and D203, on Cdc20$^{APC/C,}$ suggesting that these residues are involved in electrostatic interactions. The CRY box has a U shape with R166 performing the U-turn of the polypeptide chain (Fig. 3c). This specific arrangement is independent of the binding of MCC to the APC/C complex as it is also present in the crystal structure of Cdc20 on its own[22]. The CRY box U-turn conformation is stabilized by interactions of the conserved R166 and I168 with the first WD40 repeat of Cdc20$^{MCC}$. The CRY motif is wedged in between the acidic Cdc20$^{APC/C}$ D box receptor region and two

BubR1 interacting loops, which are juxtaposed. Because of the knot-like shape of this domain, we call it BubR1$^{KNOT}$ (Fig. 3c). The BubR1$^{KNOT}$ is formed by the BubR1 first D-box pseudo-degron sequence (residues 224–232), which entangles with the following hydrophobic loop (residues 242–252) (Fig. 3c, d). The CRY motif residues following R162 and K163 are mainly non-polar, and they establish van der Waals interactions with a non-polar region of Cdc20$^{APC/C}$ underneath the D box receptor and the hydrophobic core of the BubR1$^{KNOT}$ structure.

To investigate the function of these interactions, a series of point mutations were designed, and their effect on checkpoint was examined. The mutation includes (1) Cdc20 R162E/K163E and Cdc20 E180R/D203R; (2) Cdc20 CRY/3A and individual A mutants; (3) Cdc20 IPS/3A and individual A mutants; (4), Cdc20 CRY individual D mutants; (5) Cdc20 IPS individual D mutants; (6), BubR1 G246D/G247D/A248D (3D) and I242D/I243D/V245D/L249D (4D). Disrupting the electrostatic interaction between Cdc20$^{MCC}$ and Cdc20$^{APC/C}$ by either R162E/K163E or E180R/D203R mutations in Cdc20 significantly impaired the checkpoint (Fig. 4a). Mutations that are designed to unfold either the Cdc20$^{CRY}$ structure (CRY/3A or IPS/3A) or the BubR1$^{KNOT}$ structure (BubR1 3D or 4D) displayed strong SAC defects (Fig. 4b–d). Among the single alanine mutations, R166A alone severely impaired the checkpoint while others showed little or very mild effects (Fig. 4b). For the single aspartic acid mutations in Cdc20, R166D, Y167D, and I168D, all gave strong SAC defects further supporting a hydrophobic interaction between the CRY box and the surrounding residues (Fig. 4c). The residue Ser170 within the CRY box has been reported to be phosphorylated by Plk1 to facilitate the Cdc20 degradation in G1 phase[20]. The individual alanine or aspartic acid mutation analysis of the Ser170 suggested a marginal role of this residue and its phosphorylation in the SAC (Fig. 4b, c). Another known post-translational modification that might be involved in regulating SAC via the interaction with the CRY box is the acetylation of BubR1 K250[32]. However, using the established BubR1 RNAi and rescue method[23], we could not see a meaningful contribution of K250 acetylation in SAC activation as described before[32] (Supplementary Fig. 3c).

The above structure-function analysis indicates that multiple contacts of the Cdc20$^{MCC}$ CRY box with Cdc20$^{APC/C}$ and BubR1$^{KNOT}$ stabilize the interaction between MCC and APC/C. To confirm this, we performed immunoprecipitation in HeLa cells expressing YFP tagged wild type Cdc20, Cdc20 R166A, Cdc20 CRY/3A, wild type BubR1, BubR1 3D, BubR1 4D mutants by GFP-trap beads. Western blot analysis showed that Cdc20 R166A and CRY/3 A mutant were defective at both MCC formation and binding with APC/C. BubR1 3D or 4D mutants were still able to form MCC, though less efficiently than wild-type BubR1, and largely lost the interaction with APC/C (Fig. 4e, f). Our data also reveal that R166 is important for MCC formation, as predicted by structure analysis, which suggests that R166 may interact with and stabilize the WD40 domain. The loss of APC/C binding from R166A could be due to a combined effect of impaired MCC formation and misorientation of the CRY box. Conversely, BubR1 3D or 4D mainly affects MCC binding to APC/C$^{Cdc20}$ (Fig. 4e, f). To further confirm the above observation, we in vitro reconstituted the APC/C$^{Cdc20}$ and the MCC complexes formed with either wild-type proteins or with Cdc20 R162E/K163E or BubR1 4D point mutations since these mutants assembled in a stoichiometric complex with the other MCC subunits as the wild type proteins, showing that these mutations do not affect the structural integrity of the MCC complex in vitro (Supplementary Fig. 4). With these recombinant complexes we performed an APC/C$^{Cdc20}$ ubiquitination assay with Cyclin B1 as the substrate. The assay showed that the MCC formed by

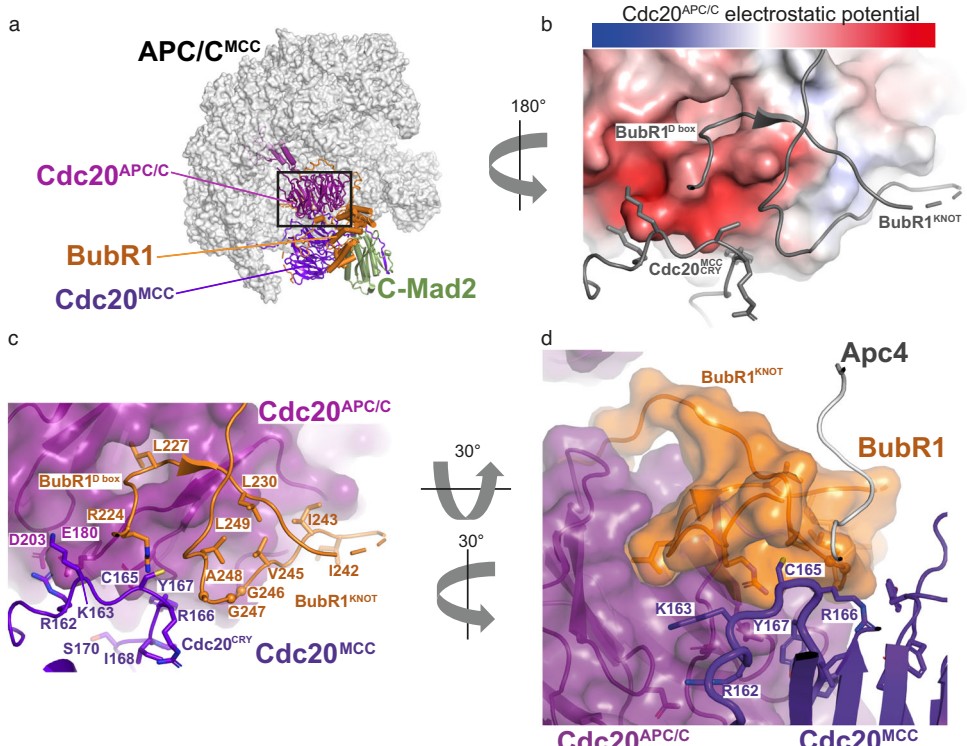

**Fig. 3 Structure analysis of the CRY motif of Cdc20$^{MCC}$ in the context of MCC-APC/C. a** Structure of the APC$^{MCC}$ complex (PDB ID: 6TLJ, Alfieri et al., 2020), the APC/C is shown as surface, the MCC subunits and Cdc20$^{APC/C}$ are shown as cartoon representation. **b** Close-up view on the Cdc20$^{MCC}$ CRY box region (from **a**), interacting with Cdc20$^{APC/C}$ and BubR1$^{KNOT}$ domain (gray cartoon). The surface electrostatic potential is shown for the Cdc20$^{APC/C}$. **c**, **d** Two close-up views showing the composite interface made of Cdc20$^{APC/C}$ and BubR1$^{KNOT}$ domain, which recognizes the Cdc20$^{MCC}$ CRY box. Cdc20$^{APC/C}$ is shown as a cartoon representation and transparent surface view in (**c**), as well as Cdc20$^{APC/C}$ and BubR1$^{KNOT}$ domain in (**d**). Cdc20$^{MCC}$ is shown as a cartoon representation in both (**c**) and (**d**). Relevant residues are depicted in (**b**–**d**).

wild-type components efficiently inhibited APC/C$^{Cdc20}$ while the MCC reconstituted with Cdc20 2E mutant or BubR1 4D mutant did not (Fig. 4g, h).

In conclusion, our functional analysis of the CRY box performed both in cells and in our in vitro reconstituted system shows that the CRY box is a critical motif within the MCC complex required for SAC-dependent inhibition of the APC/C.

**C box is exchangeable with the CRY box for APC/C and SAC activation.** The core sequence of the CRY box (CRYIPS) is highly similar to the C box (DRYIPH) as observed previously[15]. The Apc8 binding residues Arg and Ile in the C box are both present in the CRY box, and the above functional analysis revealed a certain tolerance of residue variation on the SAC of C165 or S170 within the CRY box. We reasoned that the core sequences of the two boxes could be exchanged without impairing the individual function. Cdc20 mutants with either CRY box replaced by C box or C box replaced by CRY box were constructed and examined for their function. Among wild-type Cdc20, 2 × CRY box, or 2 × C box mutants, there is no significant difference in the mitotic duration of unperturbed mitosis or SAC-activated mitosis, indicating CRY box and C box could compensate for the function of each other at their individual positions (Fig. 5a, b).

**The functionality of the Cdc20 CRY box seems conserved across eukaryotic cells.** The CRY box is well conserved from humans to zebrafish while the arginine within the box is highly conserved to worm and yeast (Fig. 5c). A similar pattern was observed for BubR1$^{KNOT}$ as well (Fig. 5d). This suggests that the

functionality of the CRY box is conserved with certain sequence variation across species (Fig. 5e).

## Discussion

In this study, we generated RNAi-sensitive cell lines expressing minimal amounts of Cdc20 by CRISPR/Cas9. How the residual Cdc20 is produced in the knockout cells is not clear. Translation reinitiation and alternative splicing have been proposed as possible mechanisms for the continued expression of the targeted gene in knockout cells[33–36]. A recent study identified two more isoforms of Cdc20 via translation reinitiation at Met 43 (M43) and Met 88 (M88)[33] which may not be the case here due to the following two reasons. First, Cdc20 KO3-9 has an extra C inserted within the codon for P74, which shifted the reading frame and induced a premature termination codon after R76 (Supplementary Fig. 1c). Thus, the full-length Cdc20 and the short isoform M43 could not be produced. M88 is defective at activating both SAC and APC/C and cannot support cell viability[33]. Secondly, none of the isoforms were detected by WB in the knockout cells (Fig. 1a; Supplementary Fig. 1d). On the other hand, a single nucleotide insertion could generate new or activate cryptic cis-element like exonic splicing silencer (ESS) or exonic splicing enhancer (ESE) on the pre-mRNA which results in alternatively spliced mRNA. Identification of such a transcript will reveal the mechanism of the production of functional Cdc20 in the knockout cells.

One interesting feature of these cells is the lack of SAC. The reason behind this is also unclear. One possibility is that the Cdc20 produced in these cells lost the motif required for MCC formation, similar to what was reported by Tsang et al.[33]. Likely

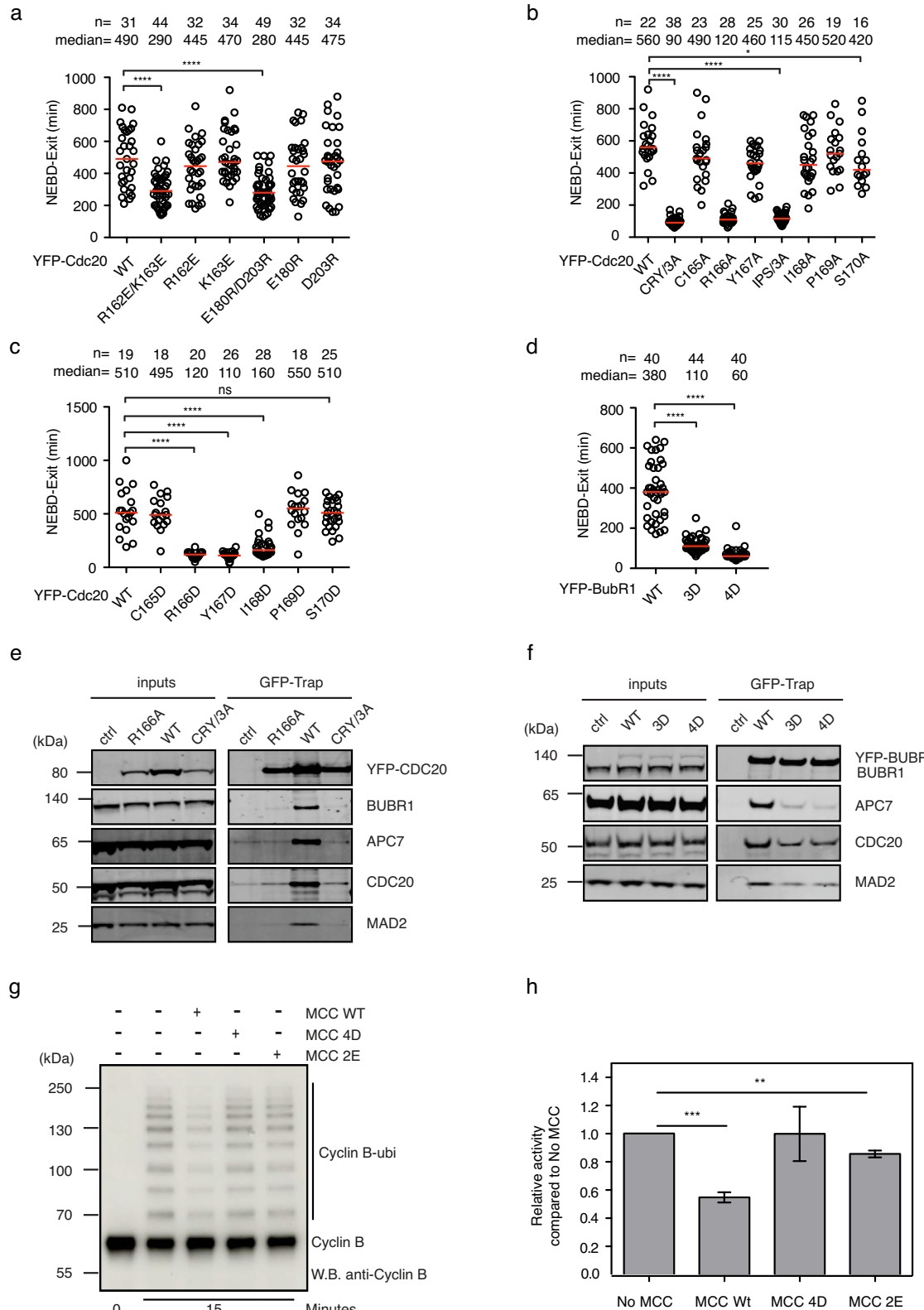

due to the very low level of Cdc20 or motif changes, APC/C can only be activated at a much lower rate. Thus, the knockout cells are provided enough time for proper attachment between kinetochores and microtubules, even in the absence of SAC. In such a scenario, the SAC is not essential anymore, which is consistent with the previous study, which found that the SAC became dispensable in cells with UBE2C and UBE2S genetically depleted[37].

Based on this system, we quantified the known motifs on Cdc20 for their contribution to SAC or APC/C activation. It has been revealed that the stable interaction between MCC and APC/C is important for ligase activity inhibition[16,38]. However, MCC-mediated APC/C inhibition requires that the APC/C could be activated by the mutated Cdc20. Whether the Cdc20 mutants used in these studies can activate APC/C within cells is not fully

**Fig. 4 Functional analysis of the interactions with CRY box. a–d** The time from NEBD to mitotic exit in the presence of nocodazole (30 ng/ml) of the knockout cells complemented with wild-type or mutant YFP-Cdc20 and depleted the residual endogenous Cdc20 by RNAi or HeLa cells complemented with wild type or mutant YFP-BubR1 and depleted endogenous BubR1 by RNAi. Each circle represents the time of a single cell, and the red line indicates the median time. The number of cells analyzed per condition is indicated above ($n = $X). A representative experiment of two independent experiments is shown. Mann–Whitney $U$-test was applied. ns means not significant; * means $P < 0.1$; **** means $P < 0.0001$. a Wild type Cdc20 or R162, K163, E180, D203 mutants were examined for SAC activity. **b** Wild-type Cdc20 or CRYIPS alanine mutants were examined for SAC activity. **c** Wild-type Cdc20 or CRYIPS aspartic acid mutants were examined for SAC activity. **d** Wild-type BubR1 or 3D, 4D mutants were examined for SAC activity. **e** Immunoprecipitation was performed using GFP-trap beads in HeLa cells expressing the corresponding YFP-Cdc20 constructs. A quantitative western blot was conducted with corresponding antibodies. A representative experiment of two independent experiments was shown. **f** Immunoprecipitation was performed using GFP-trap beads in HeLa cells expressing the corresponding YFP-BubR1 constructs. A quantitative western blot was conducted with corresponding antibodies. A representative experiment of two independent experiments was shown. g Ubiquitination of Cyclin B1 by APC/C$^{Cdc20}$ after 15 min with and without the wild type MCC or with the MCC containing Cdc20 R162E/K163E (2E) or the MCC containing BubR1 I242D/I243D/V245D/L249D (4D). See Supplementary Fig. 5 for uncropped blots in (**e–h**). Quantification of the Cyclin B1 ubiquitination assay after 15 min. The data shown is the mean value ± S.D. of 3 independent experiments. *** means $P < 0.0005$, ** means $P < 0.005$, the $P$ values were calculated using a two-tailed unpaired $t$-test.

disclosed, as the residual endogenous protein could activate APC/C to promote mitotic exit or Cyclin B1 degradation even in the presence of the Cdc20 mutants. In the clean background achieved in this study, all three Cdc20 mutants lacking C box, IR motif, or KILR motif were not able to activate APC/C and promote mitotic exit in unperturbed mitosis. Cells were arrested at metaphase till apoptosis occurred. Thus, all three APC/C binding motifs are required for the APC/C activation by Cdc20. Our data also supports the role of Cdk1 in inhibiting Cdc20 by multiple phosphorylations, which was recently questioned[39].

Regarding the MCC-APC/C binding, several interactions between components from the two complexes are involved, including the TPR domain of BubR1 with the UbcH10-binding site on Apc2$^{WHB}$, the IR tail of Cdc20$^{MCC}$ with the C box-binding domain on Apc8A, BubR1/Bub3 and phosphorylation sites on APC/C as well as the lariat-like structure between BubR1 degrons and degron receptors on the two Cdc20 molecules. In this study, multiple novel interactions have been characterized among Cdc20$^{MCC}$, Cdc20$^{APC/C}$, and BubR1. These interactions are mediated by the CRY box, which is required for both MCC formation and MCC-APC/C interaction. Individual disruption of each contact caused strong SAC defect, suggesting a critical role of the CRY box in SAC activation, and this was also confirmed by biochemical assays. So, we unveil a new layer of regulation on the MCC-mediated APC/C inhibition by the CRY box in this study. How the CRY box-mediated interactions cooperate with the other known interactions between the MCC and APC/C for the stable binding of the two protein complexes requires further investigation.

In conclusion, by using CRISPR/Cas9 and RNAi, we reanalyzed the known functional motifs on Cdc20, confirmed their essential roles in APC/C activation, and revealed one novel critical function of the CRY box on MCC inhibition of APC/C. The strategy used here could also be applied to other genes when RNAi is preferred but not able to produce a null phenotype alone.

## Materials and methods

**Cell culture, transfection**. HeLa cells were cultivated in DMEM medium (Thermo Fisher Scientific) supplemented with 10% of FBS (Biological Industries) and antibiotics. For Cdc20 RNAi and rescue experiments, cells were seeded in a 6-well plate at 50% confluence with thymidine (2.5 mM) in the medium. Twenty-four hours later, cells were released from thymidine and transfected with 450 ng of YFP-Cdc20 expression plasmid with Lipofectamine 2000 (Thermo Fisher Scientific). The next morning, 20 nM of RNAi oligos were transfected with Lipofectamine RNAiMAX (Thermo Fisher Scientific) in the presence of thymidine (2.5 mM). Cells were released from the second thymidine arrest the following morning and processed for further assays.

RNAi oligos targeting Cdc20 (5' CGGAAGACCUGCCGUUA-CAtt 3'), BubR1 (5' GAUGGUGAAUUGUGGAAUAtt 3') or luciferase (5' CGUACGCGGAAUACUUCGAtt 3') were synthesized from GenePharm.

**Cloning**. PX459 was used to generate the constructs targeting *Cdc20*. Three sequences within exon 1 and 2 selected as Cas9 targeting sites include 5' TGCAAGGACCCCTCCCCCTG 3' (#1), 5' CGCAAAGCCAAGGAAGCCGC 3' (#2) and 5' ACCACTCCTAGCAAACCTGG 3' (#3). For YFP-tagged Cdc20 expression, pcDNA5/FRT/TO N-YFP vector was double digested by restrictive enzymes of KpnI and NotI (Thermo Fisher Scientific). Wild-type *Cdc20* was amplified by PCR and inserted into the expression vector according to standard procedures. *Cdc20* mutant constructs were generated with mutation PCR. KOD DNA polymerase (Toyobo) was used for gene amplification and mutagenesis. Details of the cloning will be provided upon request.

**CRISPR/Cas9 mediated gene editing**. The construct expressing guide RNA and Cas9 protein was transfected into HeLa cells with Lipofectamine 2000. Forty-eight hours later, cells were under selection with puromycin (1 µg/ml) for two days. After puromycin selection, cells were further cultivated for two more weeks till single clones appeared. Single clones were picked and expanded for further analysis.

**Genomic DNA sequencing**. The genomic DNA of each knockout cell line was extracted with a GeneJET Genomic DNA purification kit (Thermo Fisher Scientific). Primers were designed at 200 nt upstream and downstream of the cutting site on CDC20 exons. 200 ng of genomic DNA was used as a template for the amplification of the interested region by 2× Taq Plus Master Mix (Vazyme). Amplified genomic DNA was further cloned into a pTA2 vector according to the manufacturer's instructions (TArget Clone, Toyobo). Six clones were picked and sequenced for each cell line.

**Live cell imaging**. After RNAi transfection, as described above, cells were re-seeded into chamber slide (Ibidi) from the 6-well plate with fresh DMEM medium containing thymidine. The following morning, cells were released from thymidine arrest. Five hours later, the DMEM medium was replaced by Leibovitz's L-15 medium containing 10% of FBS. For all the assays examining SAC activity, a low dose of nocodazole (30 ng/ml) was added into the medium to activate the checkpoint. The slides were mounted onto Nikon A1 HD25 confocal microscopy (Nikon). 1.68% of laser 488 nm was used for YFP signal and DIC imaging simultaneously. Signals were collected every 10 min for a

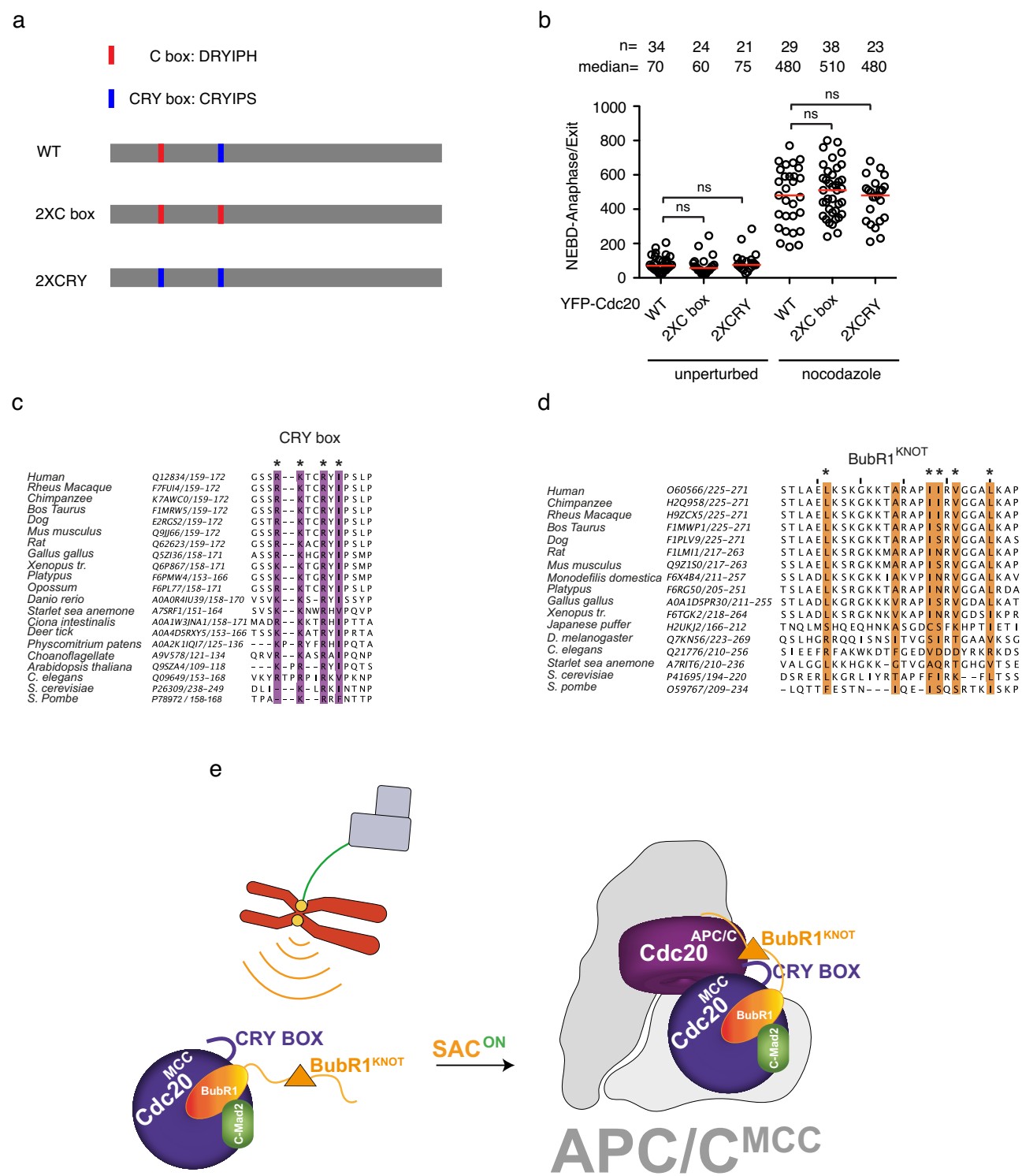

**Fig. 5 The C box and CRY box are exchangeable for their proper functioning. a** Schematic showing the design of the C box and CRY box exchanging. **b** The time from NEBD to anaphase (unperturbed) or from NEBD to mitotic exit in the presence of nocodazole (30 ng/ml) of the knockout cells complemented with wild-type or engineered YFP-Cdc20 and depleted the residual endogenous Cdc20 by RNAi. Each circle represents the time of a single cell, and the red line indicates the median time. The number of cells analyzed per condition is indicated above (*n* = X). A representative experiment of two independent experiments is shown. Mann–Whitney *U*-test was applied. ns means not significant. c,d sequence alignments showing the conservation of Cdc20 CRY BOX (**c**) and BubR1 KNOT domain (**d**). **e** Cartoon highlighting the critical role of the CRY box in mitotic checkpoint signaling. Top left: kinetochore (yellow circle) not attached by spindle microtubule (green line) organized by centrosome (gray squares) activates the spindle assembly checkpoint; Bottom left: MCC complex is generated from the unattached kinetochore. The CRY box is required for the complex formation. Right: Stable interaction between MCC and APC/C requires multiple interactions mediated by the CRY box.

total of 20 h. NIS-Elements AR Analysis was used for results analysis. To calculate the time cells spent in mitosis, the following time points were used. NEBD was recorded from the time the visible nuclear envelope dissolves. Anaphase onset was recorded as the cell elongated with visible chromosome separation. Mitotic exit upon treatment of tubulin toxin was recorded at the beginning of irregular cell division.

**Immunofluorescence**. HeLa cells were cultivated on coverslips in a 6-well plate and treated as described above. One more step of synchronization by RO3306 (5 mM) overnight treatment after the double thymidine arrest was conducted. The next morning, cells were washed twice with PBS and further cultivated for 90 min in a fresh medium containing nocodazole (200 ng/ml). Cell fixation and staining were conducted as described previously[29]. Briefly, the cells were washed once by PBS and fixed with 4% paraformaldehyde in PHEM buffer (60 mM PIPES, 25 mM HEPES, pH 6.9, 10 mM EGTA, and 4 mM $MgSO_4$) at room temperature for 20 min, followed by permeabilization with 0.5% Triton X-100/ PHEM for 10 minutes. Fixed cells were blocked by 3% BSA/PBST and stained with corresponding antibodies. The antibodies used in this study include GFP (homemade in JN lab, 1:500), CENP-C (MBL, PD030, 1:800), Mad1 (Santa Cruz, sc65494, 1:200), Mad2 (homemade in JN lab, 1:200), Bub1 (abcam, ab54893, 1:200), Cdc20 (Santa Cruz, sc13162, 1:200), BubR1 (homemade in JN lab, 1:200). Fluorophore-labeled secondary antibodies were purchased from Thermo Fisher Scientific (1:1000). ProLong Gold antifade mountant (Thermo Fisher Scientific) was used for the coverslips mounting onto slides. Z-stacks were recorded every 200 nm with a 100× oil objective and sCMOS camera (DFC9000) equipped on Thunder Imaging System (Leica). Deconvolution and kinetochore protein quantification were performed with LAS X software (Leica).

**Immunoprecipitation and Western blot**. For Cdc20 and BubR1 immunoprecipitation, 2 μg of plasmid was transfected into HeLa cells in a 15 cm dish 48 h before collection. Cells were synchronized by double thymidine arrest followed by overnight treatment by nocodazole (200 ng/ml). Mitotic cells were shaken off the plate and lysed on ice in a lysis buffer containing 10 mM Tris HCl, pH 7.4, 150 mM NaCl, 0.5 mM EDTA, and 0.5% NP40 with protease and phosphatase inhibitors (Roche). The cell lysate was centrifuged at 17,000 g for 10 min at 4 °C, and the supernatant was applied to 20 μl of GFP-Trap beads (Chromotek) and shaken for 2 h at 4 °C. The beads were washed three times with 0.5 ml lysis buffer and boiled in 50 μl 2× SDS loading buffer. For Cdc20 knocking out examination, cells from a 10 cm dish were collected and lysed in 200 μl of lysis buffer as described above. The cell lysate was cleaned by centrifugation and boiled in an SDS loading buffer. A quantitative western blot (Odyssey DLx, LI-COR) was performed to examine Cdc20 and interested proteins. Antibodies used include Cdc20 (Santa Cruz, sc-13162; Bethyl, A301-180A), BubR1 (homemade in JN lab), Mad1 (Sigma, M8069), Mad2 (homemade in JN lab), Apc7 (Santa Cruz, sc-365649), Apc15 (Santa Cruz, sc-398488) and GAPDH (Proteintech, 60004-1). Fluorophore-labeled secondary antibodies include goat anti-mouse IRDye 800CW and goat anti-rabbit IRDye 680CW (LI-COR).

**Mass spectrometry analysis**. Cells were collected by mitotic shake-off and washed twice with ice-cold PBS. Cell extracts were prepared by resuspending cell pellet in ice-cold lysis buffer (10 mM Tris-HCl, pH 7.5, 150 mM NaCl, 0.5 mM EDTA, pH 8.0, 0.5% NP-40) and incubating on ice for 30 min before centrifugation at 17,000g for 15 min at 4 °C. The supernatant was

transferred to a fresh 1.5 ml microcentrifuge tube. In total, 2 μg Cdc20 antibody (BETHYL, A301-180A) was added into 1.5 mg of cell lysate, and the mixture was rotated end-to-end at 4 °C overnight. In total, 40 μl of Protein A/G PLUS-Agarose (Santa Cruz, sc-2003) was applied to the mixture the next morning for 3 h at 4 °C on the rotating device. The beads were collected by centrifugation at 5000 rpm for 2 min at 4 °C and washed 4 times with ice-cold wash buffer (10 mM Tris/Cl pH 7.5, 150 mM NaCl, 0.5 mM EDTA). The beads were resuspended with 40 μl of 2× sample buffer and boiled for 3 min. Immunoprecipitates were separated by SDS-PAGE and sent for mass spectrometry analysis (Hangzhou Cosmos Wisdom Biotechnology).

**Protein expression and purification**
*His-Ubiquitin, His-UBA1 and UbcH10-His*. Codon-optimized cDNA of human His-ubiquitin, His-UBA1, and UbcH10-His were ordered from GenScript and Addgene and transformed into BL21(DE3)RIL cells. Large-scale bacterial cultures were shaken at 37 °C, 220 rpm until the $OD_{600}$ reached 0.6 or 0.3 for UBA1. 200 μM of IPTG was added, and the culture was continued at 18 °C, 180 rpm overnight. The culture was centrifuged at 10,000×$g$, 4 °C for 10 min. The cell pellet was resuspended in 50 mM HEPES pH 7.5, 500 mM NaCl, 5% glycerol, 0.5 mM TCEP 25 mM imidazole, 5 units/mL Benzonase, supplemented with an EDTA-free protease inhibitor cocktail (Roche). The cell suspension was sonicated and centrifuged at 40,000×$g$ for 20 min. The supernatant was loaded onto a 5 mL HisTrap-HP column (Cytiva) with a 1 mL/min flow rate. Afterward, the column was washed extensively with wash buffer (50 mM HEPES pH 7.5, 500 mM NaCl, 5% glycerol, 0.5 mM TCEP, and 25 mM imidazole), and the protein was eluted with wash buffer containing 300 mM imidazole. The elute containing His-tagged protein was pooled and concentrated before being loaded onto a HiLoad 16/ 600 Superdex 75 pg column (Cytiva) and run with gel filtration buffer (20 mM HEPES pH 7.5, 150 mM NaCl, 0.5 mM TCEP). His-UBA1 was purified as described[40].

*MBP-Cdc20, BubR1-GST, Mad2-StrepII, Cyclin B1-StrepII, APC/ C*. Codon-optimized cDNA of His-MBP-Cdc20, BubR1-GST, and Mad2-StrepII were ordered from GenScript, and Cyclin B1-StrepII was ordered from GeneArt (Thermo Fisher Scientific), which were all in pFastBac1 vectors and transformed into MultiBac cells for insect cell expression. Baculoviruses for each of these were produced in Sf9 cells and used to infect High Five cells at a cell density of $1.5 \times 10^6$ cells/mL. For expression of the MCC, the Cdc20, BubR1, and Mad2 baculoviruses were used to coinfected High Five cells (the Bub3 protein was not included because it is not required for MCC-mediated APC/C inhibition in vitro[7]). High Five cells were cultured at 27 °C, 130 rpm for 72 h.

Cdc20 was purified as described[40]. The wt and mutant MCC were all purified in the same way. The cell pellet was resuspended in 50 mM HEPES pH 7.5, 200 mM NaCl, 5% glycerol, 0.5 mM TCEP, 5 units/mL Benzonase, supplemented with an EDTA-free protease inhibitor cocktail (Roche). The cell pellets were then sonicated and centrifuged for 1 h at 55,000×$g$. The supernatant was loaded onto a 5 mL StrepTactin Superflow Plus cartridge (Qiagen) at a 1 mL/min flow rate and washed with MCC wash buffer (50 mM HEPES pH 7.5, 200 mM NaCl, 0.5 mM TCEP, 5% glycerol) and then eluted with wash buffer supplemented with 2.5 mM desthiobiotin. The fractions containing MCC were pooled and loaded onto a 5 mL GSTrap HP column (Cytiva) using a 0.5 mL/min flow rate. After extensive washing with MCC wash buffer, the MCC was then eluted with wash buffer supplemented with 10 mM glutathione. The pooled fractions were incubated on ice with TEV and HRV-3C proteases for 4 h

and then injected into a Superose 6 increase 10/300 GL (Cytiva). Fractions containing the MCC were pooled and concentrated.

The Cyclin B1 cell pellet was resuspended in 50 mM HEPES pH 8.0, 500 mM NaCl, 5% glycerol, 0.5 mM TCEP, 5 units/mL Benzonase supplemented with an EDTA-free protease inhibitor cocktail (Roche). After sonication, the lysed cells were centrifuged at 55,000×g for 1 h. The supernatant was loaded onto a 5 mL StrepTactin Superflow Plus cartridge (Qiagen) using a 1 mL/min flow rate and washed with Cyclin B1 wash buffer (50 mM HEPES pH 8.0, 500 mM NaCl, 5% glycerol, 0.5 mM TCEP) and eluted with wash buffer supplemented with 2 mM desthiobiotin. Fractions were concentrated and loaded onto a HiLoad 16/600 Superdex 200 pg column (Cytiva).

Recombinant human APC/C with a phosphomimic mutation in Apc1 was expressed in High Five insect cells. The High Five cells were infected with three separate baculoviruses, one containing Apc5, Apc8, Apc10, Apc13, Apc15, Apc2, and strep-tagged Apc4 (gifted by David Barford's lab). The second contained Apc1 phosphomimic mutant (Apc1 Ser364Glu, Ser372Glu, Ser373Glu, Ser377Glu)[41], and Apc11 and the third with Apc3, Apc6, Apc7, Apc12, and Apc16. High Five cells were infected at a cell density of $2 \times 10^6$ and cultured at 27 °C, 130 rpm for 72 h.

The APC/C cell pellet was resuspended in 50 mM HEPES pH 8.3, 250 mM NaCl, 5% glycerol, 2 mM DTT, 1 mM EDTA, 0.1 mM PMSF, 2 mM Benzamidine, 5 units/mL Benzonase supplemented with an EDTA-free protease inhibitor (Roche). Cells were sonicated and then centrifuged at 48,000×g for 1 h. The supernatant was loaded to a 5 mL StrepTactin Superflow Plus cartridge (Qiagen) using a 1 mL/min flow rate. The column was washed extensively with APC/C wash buffer (50 mM HEPES pH 8.3, 250 mM NaCl, 5% glycerol, 2 mM DTT, 1 mM EDTA, 2 mM Benzamidine) and eluted with wash buffer supplemented with 2.5 mM desthiobiotin (IBA-Lifesciences). Fractions containing APC/C were incubated overnight with tobacco etch virus (TEV) protease and then diluted two-fold with saltless Buffer A (20 mM HEPES pH 8.0, 125 mM NaCl, 5% glycerol, 2 mM DTT, 1 mm EDTA). This was loaded onto a 6 mL ResourceQ anion-exchange column (GE Healthcare), and the column was washed with buffer A. The APC/C was eluted with a gradient of Buffer B (20 mM HEPES pH 8.0, 1 M NaCl, 5% glycerol, 2 mM DTT, 1 mM EDTA). Concentrated APC/C was centrifuged (Optima TLX Ultracentrifuge) at 40,000 rpm for 30 min.

**Ubiquitination assay.** 50 nM APC/C, 50 nM Cdc20, 200 nM Cyclin B1, 30 nM UBA1, 20 μM Ubiquitin, 5 mM ATP, 0.25 mg/mL BSA and 250 nM UbcH10 were mixed in a 15 μL reaction volume with reaction buffer: 40 mM HEPES pH 8.0, 0.6 mM DTT, 10 mM MgCl₂. 270 nM of each MCC construct was added into its respective reaction. The reaction mixture was incubated at 25 °C for 15 minutes. Samples were taken at 0 and 15 min, and the reaction was terminated with SDS/PAGE loading dye. The reaction mixture was run on a 4–12% NuPAGE Bis-Tris gel and transferred to a nitrocellulose membrane for Western blotting. Blocking was carried out in 5% BSA-TBST and washed in 1× TBST. Ubiquitin-modified Cyclin B1 was detected using an anti-CyclinB1 mAb (ABclonal, A19037) at a 1:2,000 dilution, and HRP-conjugated goat anti-rabbit secondary antibody (Abcam, ab205718) at a 1:10,000 dilution. The membrane was incubated with ECL, detected using the ImageQuant 800 (Amersham), and quantified using ImageJ.

**Statistics and reproducibility.** The Mann–Whitney U-test was applied for all the statistic analyses for live imaging results, which have been repeated at least twice. ns means not significant; *

means $P < 0.1$; ** means $P < 0.01$; *** means $P < 0.001$; **** means $P < 0.0001$. For ubiquitination assay, the $P$ values were calculated using a two-tailed unpaired $t$-test. The data shown is the mean value ± S.D. of 3 independent experiments. *** means $P < 0.0005$, ** means $P < 0.005$.

**Reporting summary.** Further information on research design is available in the Nature Portfolio Reporting Summary linked to this article.

## Data availability

All data supporting the findings of this study are available within the paper and its supplementary information, except the mass spectrometry proteomics data, which have been deposited to the ProteomeXchange Consortium via the PRIDE partner repository with the access code PXD045411. Source data can be found in Supplementary Data 1.

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

## Acknowledgements

G.Z. is supported by the National Natural Science Foundation of China (31970666) and the Taishan Scholar Project (tsqn201812054) from Shandong, China. C.A. and R.Y. are supported by the Sir Henry Dale Fellowship 215458/Z/19/Z. R.Y. is also supported by the Institute of Cancer Research (ICR), and the grant number allocated is GFR005X. We acknowledge Jing Yang, Ziguo Zhang, and David Barford for helping with the APC/C baculovirus generation.

## Author contributions

YZ (Yuqing Zhang) conducted all the cloning, cell line generation and function analysis. RY purified the recombinant protein complexes and did a ubiquitination assay. DHG did Cdc20 and BubR1 immunoprecipitation. CS, YZ (Yujing Zhai), YW, HJ assisted YZ (Yuqing Zhang) on data collection and analysis. JF provided constructive advice for the project. JN wrote the paper together with CA and GZ. CA did the structural analysis and supervised the whole project with GZ.

## Competing interests

The authors declare no competing interests.
