## [Peer Review File · Communications Biology]

Reviewers' comments:

Reviewer #1 (Remarks to the Author):

The manuscript titled "Functional Analysis of Cdc20 Reveals a Critical Role of CRY Box in Mitotic Checkpoint Signaling" by Zhang and colleagues conducts a comprehensive functional analysis of Cdc20, aiming to dissect the contributions of its known motifs to protein functions. With Cdc20 playing a pivotal role in cell cycle regulation—serving as an activator of the Anaphase-Promoting Complex (APC) for proper chromosome segregation and being integral to the Mitotic Checkpoint Complex (MCC) to prevent premature APC activation—the study is both intriguing and relevant.

The authors employ a well-established CRISPR/Cas9 technology-based protocol, previously successful in studying Bub1, to engineer cell lines. Among various notable observations, the study unveils a critical function of the poorly understood CRY box in inhibiting APC/C within the MCC, revealing a novel layer of regulation.

While the study is thorough and well-executed, a notable critique lies in the manuscript's writing style. The text appears highly technical and lacks clear descriptions of rationale and logical explanations, making it challenging for a broader audience to comprehend. Throughout the manuscript, clarity in the presentation of logic, rationale, and experimental design is lacking, necessitating additional reading for full understanding.

In addition:

- The manuscript lacks information on statistical analyses and sample sizes, which is important for ensuring the robustness of the study's findings. Inclusion of such details would enhance the transparency and completeness of the manuscript.
- It is not clear how the authors assess the time from NEBD to anaphase or exit. The manuscript should provide clarification on their specific readout.
- On page 5, when referring to YFP-Cdc20 Fig.1C, it should be corrected to Fig.1G.
- Supplementary Fig.1 should be reorganized for clarity to enhance its accessibility to readers.

Reviewer #2 (Remarks to the Author):

Zhang et al. investigated the function of Cdc20, an activator of APC/C, as well as a component of the MCC complex at the spindle assembly checkpoint (SAC). While previous research has been hindered by the difficulty of conditionally inactivating endogenous Cdc20, in this manuscript, the authors first demonstrate a method to overcome this challenge by combining CRISPR-Cas9 and siRNA depletion. Using their novel method, they investigate the impacts of mutations in several important motifs in Cdc20 during mitosis. They not only confirm the significance of several motifs in mitosis but also discover that the CRY box, previously known for its role in Cdc20 degradation, plays a crucial role in the MCC-driven inactivation of APC/C. This is achieved through protein-protein interaction, forming a stable inhibitory complex.

Overall, the presented data are clear, and the authors' interpretations are rational. Their establishment of a method to analyze Cdc20 mutants in cells, along with their discovery of a novel role for the CRY box, marks significant progress in our understanding of the molecular mechanisms that support the inactivation of APC/C by MCC. However, I believe there are several points the authors should address before this paper is ready for publication.

Major Comments:

- My primary concern is the ambiguity surrounding why cells with a Cdc20 knock-out can survive in this system, and why they demonstrate heightened sensitivity to siRNA. As the authors pointed out in the main text, it's reported that active Cdc20 isoforms can be expressed in cells and that these isoforms can drive mitotic exit in the absence of full-length Cdc20. Therefore, the authors should examine and report the expression levels of Cdc20 isoforms. If these isoforms can account for the survival of Cdc20-knock out cells, it would clarify a lot. If not, the data would still provide valuable context to help readers better evaluate the presented results.
- Besides survival, the generated RNAi-sensitive Cdc20 knock-out cell lines lack the SAC. For example, is Cdh1 overexpressed when Cdc20 is knocked out? Could Cdh1 play a role in the survival and SAC deficiency of these Cdc20 knock-out cells? If not, what is the likely explanation for SAC deficiency?

Minor Comments:

- For Fig.1G, a description of how this experiment was performed should be added to either the main text or the figure legend.
- In Fig.2C, since nocodazole was used at varying concentrations, it would be beneficial for the readers if the specific concentrations were mentioned in the figure legend.
- In Fig. 4GH, I wondered why the authors chose Cdc20 R162E/K163E over Cdc20 R166A or CRY/3A, especially given the results in Fig4E. It raises the question if Cdc20 R166A and CRY/3A might not reconstitute functional MCC to inhibit APC/C and if Cdc20 R162E/K163E can form MCC, similar to the tests in Fig4E. Consistent data between in vivo and in vitro experiments would strengthen the manuscript.
- For Supp Fig 1B, it would be beneficial to include the position of the siRNA-targeting sequence and any potential isoforms.
- In Supp Fig 1C, the current presentation makes it difficult to identify the introduced mutations and their subsequent impact on Cdc20 expression. It would be helpful if this information was highlighted within the figure.
- Lastly, there seem to be recurring typographical errors throughout the manuscript and several figures. For instance, the term "medium" should likely be corrected to "median."

Reviewer #3 (Remarks to the Author):

The timing of anaphase onset is a critical determinant of mitotic chromosome segregation fidelity, and the protein Cdc20 plays a key role in governing anaphase onset. Although Cdc20 has been the subject of numerous investigations, its multi-faceted N-terminus still retains some mystery. Zhang and colleagues use a carefully designed methodology to define the specific roles of the protein-protein interaction motifs that reside within the Cdc20 N-terminus. Their results unambiguously confirm prior findings and provide new insight into the role of 'CRY' box of Cdc20 in mediating its interactions with the APC/C (Anaphase Promoting Complex/Cyclosome) and BubR1 (a component of the mitotic checkpoint complex).

This study is very well designed, rigorously executed, and clearly presented. It presents novel insights into the structural basis and the mitotic role of a conserved activity of Cdc20. Therefore, this study will be very useful for the cell cycle field. Therefore, I enthusiastically support its publication with only minor modifications.

1. The word "median" appears to have morphed into 'medium' in many places in the text and figures.
2. Figure 3 is critical to the paper because it presents the structure-based hypothesis formulated by the authors. However, I found it hard to decipher in the current form. The authors should consider the following revisions.

- (a) In panel B, show the electrostatic potential surfaces for the MCC-Cdc20 and APC/C-Cdc20.
 - (b) In panel C, only show cartoon representations using different colors for the two complexes and hiding extraneous regions of APC/C (shown currently).
 - (c) Explain the relationship between the viewpoints in panels C and D.
3. The BubR1-4D mutant appears to be affecting MCC formation also in addition to the MCC-APC/C interaction (Figure 4F). The authors should comment on this as appropriate.

Reviewers' comments:

Reviewer #1 (Remarks to the Author):

The manuscript titled "Functional Analysis of Cdc20 Reveals a Critical Role of CRY Box in Mitotic Checkpoint Signaling" by Zhang and colleagues conducts a comprehensive functional analysis of Cdc20, aiming to dissect the contributions of its known motifs to protein functions. With Cdc20 playing a pivotal role in cell cycle regulation—serving as an activator of the Anaphase-Promoting Complex (APC) for proper chromosome segregation and being integral to the Mitotic Checkpoint Complex (MCC) to prevent premature APC activation—the study is both intriguing and relevant.

The authors employ a well-established CRISPR/Cas9 technology-based protocol, previously successful in studying Bub1, to engineer cell lines. Among various notable observations, the study unveils a critical function of the poorly understood CRY box in inhibiting APC/C within the MCC, revealing a novel layer of regulation.

While the study is thorough and well-executed, a notable critique lies in the manuscript's writing style. The text appears highly technical and lacks clear descriptions of rationale and logical explanations, making it challenging for a broader audience to comprehend. Throughout the manuscript, clarity in the presentation of logic, rationale, and experimental design is lacking, necessitating additional reading for full understanding.

We thank for the reviewer's constructive comments. We now revised the manuscript with more background information in the introduction, results and discussion which makes the whole manuscript more accessible to general readers.

In addition:

- The manuscript lacks information on statistical analyses and sample sizes, which is important for ensuring the robustness of the study's findings. Inclusion of such details would enhance the transparency and completeness of the manuscript.

Now we updated the sample size and statistical analysis.

- It is not clear how the authors assess the time from NEBD to anaphase or exit. The manuscript should provide clarification on their specific readout.

The definition of NEBD to anaphase or mitotic exit has been updated in the figure legends and methods now.

- On page 5, when referring to YFP-Cdc20 Fig.1C, it should be corrected to Fig.1G.

We apologize for the confusion caused by putting Fig 1C together with Fig 2C-F. The original idea was to show that knockout cells are arrested at metaphase by removing the residual Cdc20 by RNAi

(Fig. 1C). Reintroducing RNAi-resistant YFP-Cdc20 restored the normal mitotic progression (Fig. 2C-F). Now we moved Fig.1C to the correct position.

- Supplementary Fig.1 should be reorganized for clarity to enhance its accessibility to readers.

We now added the targeted positions of each gRNA in Supplementary Fig 1B and marked the indels and their consequences on the reading frame in Supplementary Fig 1C, which is now easier for the readers to access these data.

Reviewer #2 (Remarks to the Author):

Zhang et al. investigated the function of Cdc20, an activator of APC/C, as well as a component of the MCC complex at the spindle assembly checkpoint (SAC). While previous research has been hindered by the difficulty of conditionally inactivating endogenous Cdc20, in this manuscript, the authors first demonstrate a method to overcome this challenge by combining CRISPR-Cas9 and siRNA depletion. Using their novel method, they investigate the impacts of mutations in several important motifs in Cdc20 during mitosis. They not only confirm the significance of several motifs in mitosis but also discover that the CRY box, previously known for its role in Cdc20 degradation, plays a crucial role in the MCC-driven inactivation of APC/C. This is achieved through protein-protein interaction, forming a stable inhibitory complex.

Overall, the presented data are clear, and the authors' interpretations are rational. Their establishment of a method to analyze Cdc20 mutants in cells, along with their discovery of a novel role for the CRY box, marks significant progress in our understanding of the molecular mechanisms that support the inactivation of APC/C by MCC. However, I believe there are several points the authors should address before this paper is ready for publication.

Major Comments:

- My primary concern is the ambiguity surrounding why cells with a Cdc20 knock-out can survive in this system, and why they demonstrate heightened sensitivity to siRNA. As the authors pointed out in the main text, it's reported that active Cdc20 isoforms can be expressed in cells and that these isoforms can drive mitotic exit in the absence of full-length Cdc20. Therefore, the authors should examine and report the expression levels of Cdc20 isoforms. If these isoforms can account for the survival of Cdc20-knock out cells, it would clarify a lot. If not, the data would still provide valuable context to help readers better evaluate the presented results.

We thank the reviewer for the constructive comments and apologize for the insufficient description of the rationale and the possible mechanisms. The original idea of combining CRISPR with RNAi came from our previous work on Bub1 (Zhang et al., 2019). In 2018, several labs including ours reported that BUB1 knockout cells maintained almost full SAC activity. We found these cells to be very sensitive to Bub1 RNAi indicting the continued expression of residual Bub1. By using mass spectrometry, we successfully detected Bub1 peptides in the reported knockout cells which confirmed our hypothesis.

In this study, we believe the Cdc20 knockout cells also express residual Cdc20 to support cell viability. The mechanisms for residual protein of the targeted gene have been proposed to include

translation reinitiation and alternative splicing (Anderson et al., 2017; Smits et al., 2019; Tuladhar et al., 2019; Tsang and Cheeseman, 2023). Recently a study from the Cheeseman lab identified two isoforms of Cdc20 which rely on translation reinitiation at Met 43 (M43) and Met 88 (M88). However, we don't think it is the short isoform M43 or M88 which cause the loss of SAC and support cell viability in our case due to the following reasons. First, Tsang and Cheeseman found that disrupting the first ATG only abolished the expression of full length of Cdc20, but not M43 and M88. In our knockout cells generated by guide 2 and guide 3, none of the three isoforms could be detected by WB anymore (in Fig 1C and updated Supplementary Fig 1D, attached here as figure A). Second, using KO3-9 cell as an example, an extra C was inserted within the codon for P74 at the Cas9 cutting site which shifted the reading frame and induced a premature termination codon after R76*. Thus, the full length isoform and M43 could not be produced. The third isoform M88 was defective at activating SAC and APC/C due to the loss of BM1 (27-34) and C box (77-83) and it could not support the mitotic progression (Tsang and Cheeseman, 2023).

Based on the above reasons, we believe it is more likely that an aberrant Cdc20 arising from alternative splicing may play a role here. The insertion of C may affect the correct recognition of the boundary between intron 1 and exon 2, generating a novel or activate cryptic cis-element at the pre-mRNA, such as exonic splicing silencer (ESS) or exonic splicing enhancer (ESE). The results could be a mRNA with partial deletion of exon 2 or partial retention of intron 1. The lack of SAC could be due to the loss of certain motifs required for MCC formation at the N-terminal region of Cdc20. Meanwhile, the motif changes may also partially reduce its activation of the APC/C or that very low level of protein activates the APC/C at much lower rate, so the knockout cells have enough time to generate stable kinetochore-microtubule attachment and segregate the replicated sister chromatids correctly even in the absence of SAC. Another non-exclusive possibility is that all the residual Cdc20 binds APC/C before and during mitosis. No available pool exists for MCC formation. We tried to isolate the mRNA for the residual Cdc20 from the knockout cells, but failed as the full length Cdc20 transcript with indels was still the dominant form. We expand the potential mechanisms in the discussion now.

Smits AH, Ziebell F, Joberty G, Zinn N, Mueller WF, Clauder-Münster S et al. Biological plasticity rescues target activity in CRISPR knock outs. *Nat Methods*. 2019;16(11):1087-93.

Tuladhar R, Yeu Y, Piazza, JT, Tan Z, Clemenceau JR, Wu X et al. CRISPR-Cas9-based mutagenesis frequently provokes on-target mRNA misregulation. *Nat Commun*. 2019;10(1):4056.

Anderson JL, Mulligan TS, Shen M-C, Wang H, Scahill CM, Tan FJ et al. mRNA processing in mutant zebrafish lines generated by chemical and CRISPR-mediated mutagenesis produces unexpected transcripts that escape nonsense-mediated decay. *PLoS Genet*. 2017;13(11):e1007105.

Tsang MJ, Cheeseman IM. Alternative CDC20 translational isoforms tune mitotic arrest duration. *Nature*. 2023;617(7959):154-61.

- Besides survival, the generated RNAi-sensitive Cdc20 knock-out cell lines lack the SAC. For example, is Cdh1 overexpressed when Cdc20 is knocked out? Could Cdh1 play a role in the survival and SAC deficiency of these Cdc20 knock-out cells? If not, what is the likely explanation for SAC deficiency?

We checked the expression level of Cdh1 but did not find enhanced expression in the knockout cells (see attached figure B). The lack of SAC may be caused by several possible reasons as explained above.

Minor Comments:

- For Fig.1G, a description of how this experiment was performed should be added to either the main text or the figure legend.

It is now provided in the figure legend.

- In Fig.2C, since nocodazole was used at varying concentrations, it would be beneficial for the readers if the specific concentrations were mentioned in the figure legend.

The specific concentration of nocodazole used in each assay is now specified in figure legends and in the methods.

- In Fig. 4GH, I wondered why the authors chose Cdc20 R162E/K163E over Cdc20 R166A or CRY/3A, especially given the results in Fig4E. It raises the question if Cdc20 R166A and CRY/3A might not reconstitute functional MCC to inhibit APC/C and if Cdc20 R162E/K163E can form MCC, similar to the tests in Fig4E. Consistent data between in vivo and in vitro experiments would strengthen the manuscript.

This is a very good point raised by the reviewer, we have added the following sentence clarifying this (lane 233-239): “To further confirm the above observation, we *in vitro* reconstituted the APC/C^{Cdc20} and the MCC complexes formed with either wild type proteins or with Cdc20 R162E/K163E or BubR1 4D point mutations, since these mutants assembled in a stoichiometric complex with the other MCC subunits as the wild type proteins, showing that these mutations do not affect the structural integrity of the MCC complex *in vitro* (Supplementary Fig. 4).”

- For Supp Fig 1B, it would be beneficial to include the position of the siRNA-targeting sequence and any potential isoforms.

The nucleotide position on the genomic DNA recognized by each guide RNA was listed below the scheme now.

- In Supp Fig 1C, the current presentation makes it difficult to identify the introduced mutations and their subsequent impact on Cdc20 expression. It would be helpful if this information was highlighted within the figure.

We thank the reviewer for the advice. The position of indels and the effects on amino acids have been provided now.

- Lastly, there seem to be recurring typographical errors throughout the manuscript and several figures. For instance, the term "medium" should likely be corrected to "median."

The error has been corrected now.

Reviewer #3 (Remarks to the Author):

The timing of anaphase onset is a critical determinant of mitotic chromosome segregation fidelity, and the protein Cdc20 plays a key role in governing anaphase onset. Although Cdc20 has been the subject of numerous investigations, its multi-faceted N-terminus still retains some mystery. Zhang and colleagues use a carefully designed methodology to define the specific roles of the protein-protein interaction motifs that reside within the Cdc20 N-terminus. Their results unambiguously confirm prior findings and provide new insight into the role of 'CRY' box of Cdc20 in mediating its interactions with the APC/C (Anaphase Promoting Complex/Cyclosome) and BubR1 (a component of the mitotic checkpoint complex).

This study is very well designed, rigorously executed, and clearly presented. It presents novel insights into the structural basis and the mitotic role of a conserved activity of Cdc20. Therefore, this study will be very useful for the cell cycle field. Therefore, I enthusiastically support its publication with only minor modifications.

1. The word "median" appears to have morphed into 'medium' in many places in the text and figures.

We thank the reviewer's help on identifying the typing error. It has been corrected now.

2. Figure 3 is critical to the paper because it presents the structure-based hypothesis formulated by the authors. However, I found it hard to decipher in the current form. The authors should consider the following revisions.

(a) In panel B, show the electrostatic potential surfaces for the MCC-Cdc20 and APC/C-Cdc20.

- (b) In panel C, only show cartoon representations using different colors for the two complexes and hiding extraneous regions of APC/C (shown currently).
- (c) Explain the relationship between the viewpoints in panels C and D.

We have modified the figures according to the reviewer's comments.

3. The BubR1-4D mutant appears to be affecting MCC formation also in addition to the MCC-APC/C interaction (Figure 4F). The authors should comment on this as appropriate.

It is true that BubR1-4D reduces the MCC formation in cells although this is not the case *in vitro* with recombinantly expressed proteins (Supplementary Fig. 4B,C and D). To clarify this we added the following sentence to our manuscript (line 233-239): “To further confirm the above observation, we *in vitro* reconstituted the APC/C^{Cdc20} and the MCC complexes formed with either wild type proteins or with Cdc20 R162E/K163E or BubR1 4D point mutations, since these mutants assembled in a stoichiometric complex with the other MCC subunits as the wild type proteins, showing that these mutations do not affect the structural integrity of the MCC complex *in vitro* (Supplementary Fig. 4).”

REVIEWERS' COMMENTS:

Reviewer #1 (Remarks to the Author):

The manuscript "Functional Analysis of Cdc20 Reveals a Critical Role of CRY Box in Mitotic Checkpoint Signaling" by Zhang and colleagues explores the functional aspects of Cdc20, a key player in cell cycle regulation. Through a CRISPR/Cas9 technology-based approach, the authors engineer cell lines and uncover a crucial role of the CRY box in inhibiting APC/C within the Mitotic Checkpoint Complex (MCC). This discovery introduces a novel layer of regulation to Cdc20, providing insights into previously unknown mechanisms in mitotic checkpoint signaling.

In the revised manuscript, the authors have addressed most criticisms from reviewers. While this reviewer is satisfied with the updated version, it is acknowledged that further improvement is needed in English and punctuation throughout the manuscript and, particularly, in a few sentences that remain cryptic and confusing.

As an example:

1. It would help the reader to spell each acronyms the first time it is used in each individual section (Introduction; results, etc)
2. Line 47: "besides activating APC" should read "besides activating THE APC"
3. Sentence from Line 78 to 80 is unclear

Reviewer #2 (Remarks to the Author):

The authors have addressed my concerns, either textually or through new experiments. Their efforts have significantly improved the manuscript. I am happy to support the publication of this revised version.

Reviewer #3 (Remarks to the Author):

The authors have satisfactorily addressed my comments and those of the others as far as I can see. I recommend its publication.

Reviewer #1 (Remarks to the Author):

The manuscript "Functional Analysis of Cdc20 Reveals a Critical Role of CRY Box in Mitotic Checkpoint Signaling" by Zhang and colleagues explores the functional aspects of Cdc20, a key player in cell cycle regulation. Through a CRISPR/Cas9 technology-based approach, the authors engineer cell lines and uncover a crucial role of the CRY box in inhibiting APC/C within the Mitotic Checkpoint Complex (MCC). This discovery introduces a novel layer of regulation to Cdc20, providing insights into previously unknown mechanisms in mitotic checkpoint signaling.

In the revised manuscript, the authors have addressed most criticisms from reviewers. While this reviewer is satisfied with the updated version, it is acknowledged that further improvement is needed in English and punctuation throughout the manuscript and, particularly, in a few sentences that remain cryptic and confusing.

As an example:

1. It would help the reader to spell each acronyms the first time it is used in each individual section (Introduction; results, etc)
2. Line 47: "besides activating APC" should read "besides activating THE APC"
3. Sentence from Line 78 to 80 is unclear

We thank the reviewer for the suggestions.

1, We now made several changes of the acronyms like RNAi into RNA interference (line 29), sgRNA into single-guide RNA (sgRNA) (line 94), siRNA into small interfering RNA (siRNA).

2, We added THE in line 47.

3, We modified the sentence from line 78 to 80 from "we found the disruption of any one of the three APC/C binding motifs on Cdc20 abolished the activation of APC/C" into "we found the disruption of the APC/C binding motifs individually on Cdc20 completely abolished the activation of APC/C".

Reviewer #2 (Remarks to the Author):

The authors have addressed my concerns, either textually or through new experiments. Their efforts have significantly improved the manuscript. I am happy to support the publication of this revised version.

We thank the reviewer for the support.

Reviewer #3 (Remarks to the Author):

The authors have satisfactorily addressed my comments and those of the others as far as I can see. I recommend its publication.

We thank the reviewer for the support.